# Distinct Responses to Light in Plants

**DOI:** 10.3390/plants9070894

**Published:** 2020-07-15

**Authors:** Rita Teresa Teixeira

**Affiliations:** BioISI—Biosystems & Integrative Sciences Institute, Faculty of Sciences, University of Lisbon, 1749-016 Lisbon, Portugal; rtteixeira@fc.ul.pt

**Keywords:** light sensing, photosynthesis, photoreceptors, seasonal development, photomorphogenesis, flowering control, dormancy, skotomorphogenesis

## Abstract

The development of almost every living organism is, to some extent, regulated by light. When discussing light regulation on biological systems, one is referring to the sun that has long been positioned in the center of the solar system. Through light regulation, all life forms have evolved around the presence of the sun. As soon our planet started to develop an atmospheric shield against most of the detrimental solar UV rays, life invaded land, and in the presence of water, it thrived. Especially for plants, light (solar radiation) is the source of energy that controls a high number of developmental aspects of growth, a process called photomorphogenesis. Once hypocotyls reach soil′s surface, its elongation deaccelerates, and the photosynthetic apparatus is established for an autotrophic growth due to the presence of light. Plants can sense light intensities, light quality, light direction, and light duration through photoreceptors that accurately detect alterations in the spectral composition (UV-B to far-red) and are located throughout the plant. The most well-known mechanism promoted by light occurring on plants is photosynthesis, which converts light energy into carbohydrates. Plants also use light to signal the beginning/end of key developmental processes such as the transition to flowering and dormancy. These two processes are particularly important for plant´s yield, since transition to flowering reduces the duration of the vegetative stage, and for plants growing under temperate or boreal climates, dormancy leads to a complete growth arrest. Understanding how light affects these processes enables plant breeders to produce crops which are able to retard the transition to flowering and avoid dormancy, increasing the yield of the plant.

## 1. Overview of Light Transduction Scheme

All life is shaped by the Earth’s rotation and orbit around its own axis and the sun, providing a diurnal rhythm of day and night, and depending on the latitude, an annual cycle of day length which reaches its maximum and minimum on the solstices of summer and winter, respectively. Plants use dawn and dusk as signals to organize all aspects of their growth, such as photosynthesis. Light is an environmental factor permanently present, and yet displays a dynamic control on components of plant’s functionality such as germination, phototropism, and reproduction. The decreasing number of hours of light, which starts right after the beginning of autumn, also works as an indicator to enter dormancy, preventing the detrimental effects that winter conditions might have on plant cells. Plants also react to variations of low and high light intensities (e.g., acclimation) [1], which involves a set of environmental adjustments such as leaf morphology variations, chloroplast structure, and/or modification of the composition of the photosynthetic electron transport chain, consequently affecting photosynthesis [2,3]. Plants growing under limiting light conditions show leaf and chloroplast movements toward the source of light, as well as general bending to maximize light capture, increasing photosynthetic efficiency. On the other hand, when plants are submitted to environmental conditions of high light intensity and UV-B, the chloroplasts and nucleus move away from the light, and protective photoreceptors pigments are synthesized to prevent photodamage [4,5]. These two types of acclimation responses can be reversible (dynamic acclimation) or can become permanent, resulting in irreversible or developmental acclimation mechanisms adopted by the plant [6,7].

The primary determinant of crop yield is the cumulative rate of photosynthesis over the growing season, which is regulated by the amount of light captured and its ability to efficiently use energy to convert CO_2_ into biomass and harvestable yield stored as carbohydrates [8]. The use of light during photosynthesis is achieved through two photosystems present in the chloroplast’s photosynthetic pigments, which show a narrow pick absorption range of the solar spectrum despite pigments absorbed throughout the photosynthetically active radiation (PAR) spectrum. Unlike the light regulation mediated by photoreceptor genes present in the nucleus and responsible for physiological responses such as germination, flowering, circadian clock input, and dormancy, photosynthesis requires a coordinated regulation of the nuclear and plastidic genomes [9]. In the chloroplast, photons of light are converted into glucose molecules in a two-step reaction pathway. First, energy is stored in the bonds of adenosine triphosphate (ATP) and nicotinamide adenine dinucleotide phosphate (NADPH), which are then used in the Calvin cycle (second step) to produce organic molecules after combining carbon atoms from carbon dioxide (CO_2_) with glucose as the end result. This six-carbon molecule is then utilized by the mitochondria to produce ATP. Diurnal cycle induces two distinct photosynthetic reactions: The light reaction, where water molecules are split into oxygen, hydrogen, protons, and electrons; and the dark reaction, where protons and electrons are taken to reduce CO_2_ to carbohydrates [10]. This way, photosynthesis is the primary source of the food chain because it promotes the conversion of solar energy into chemical energy.

Plants are able to utilize their internal circadian clock to synchronize physiology and development according to daily and yearly environmental changes and season cycles [11]. The circadian clock comprises a substantial number of gene activations and inhibitions in multiple feedback regulations, which are focused on two transcription factors containing DNA-binding motifs MYB: CIRCADIAN CLOCK-ASSOCIATED 1 (CCA1) and LATE ELONGATED HYPOCOTYL (LHY) [12]. These proteins activate the transcription of *PSEUDO-RESPONSE REGULATOR 9* (*PRR9*) and *PRR7* in the morning [13]. The diurnal cascade of gene activation in Arabidopsis initiates at dawn with a pick of *CCA1* transcripts. Throughout the day, *PRR9*, *PRR7*, and *PRR5* spike up early in the morning, mid-day, and afternoon, respectively. The feedback activation of these genes starts with the binding of CCA1 to the *PRR9* and *PRR7* promoter region, with further activation of these genes. In turn, PRR9 and PRR7 proteins bind to the *CCA1* promoter, inhibiting its transcription during the day [14]. Alternatively, CCA1 and LHY inhibit the expression of other clock genes that accumulate in the afternoon and evening, such as *PRR5*, *TIMING OF CAB EXPRESSION 1* (*TOC1*), *CCA1 HIKING EXPEDITION* (*CHE*), *GIGANTEA* (*GI*), *LUX ARRHYTHMO* (*LUX*), and *EARLY FLOWERING 4* (*ELF4*) [15,16]. As night approaches, CCA1 and LHY protein levels decrease, and its repressing action no longer takes place on its target genes, leading to the accumulation of the evening clock gene transcripts [17]. Besides the regulation of *PRR7* and *PRR5*, still during the day, CCA1 and LHY also regulate the expression timing of *GI* and *FLAVIN-BINDING KELCH REPEAT F-Box 1* (*FKF1*), all of which regulate *CONSTANT* (*CO*) expression [18,19]. CO is an important integration protein in the control of flowering in the correct season [20]. The PRRs also regulate the expression of the *CO* repressor, *CYCLING DOF FACTOR 1* (*CDF1*) [19]. In turn, its degradation is mediated by the GI-CDF complex with the subsequent activation of *CO* [21]. Another circadian clock feedback activation mechanism takes place during the morning, when CDF1 with other CDFs and FLOWERING BHLHs (FBHs) repress *CO* expression. The FKF1-GI complex degrades CDFs, facilitating the expression of *CO* [22]. The GI and ZEITLUPE (ZTL) complex are also responsible for TOC1 degradation by the 26S proteasome upon blue light and temperature sensing [23]. All these complex regulatory mechanisms permit the precise expression of the florigen *FLOWERING LOCUS T* (*FT*), which is responsible for flowering. For example, in Arabidopsis, a long-day (LD) responsive plant, flowering is promoted by CO, which, in turn, activates *FT* under LD [24].

Fully expanded leaves, which play an important role in capturing light, are also important for flowering, as *FT* is expressed in specific phloem companion cells in this organ [25]. The florigen FT is mobile, travelling from leaves to the SAM, where it promotes flowering in Arabidopsis [26] through the initiation of the expression of floral identity genes by binding to the bZIP transcription factor *FD* [27]. In the tree Populus, however, it is possible to identify two *FT* homologues acting in different pathways: *FT1* is involved in the reproductive onset with its transcript picking under short days (winter time), whereas *FT2* is involved in vegetative growth and cessation or bud set (dormancy) in response to seasonal day length changes [28]. Trees growing under temperate and boreal climates must adapt their growth in order to avoid the detrimental effects that winter cold might inflict on cells. This way, as day lengths become shorter, *FT2* expression is reduced, and both apical and radial growth are arrested [29]. FT2 protein is an important integrator of light regulation which controls vegetative growth under LD. The transcription of *FT2* takes place upon CO accumulation, which is mediated by the activation of phytochromes [30]. Notably, the required *FT2* transcription threshold, which is achieved in long days, coincides with the higher photosynthetic rate, resulting in the necessary carbon supply for active growth.

Plants make use of a vast number of types of photoreceptors so they can accurately detect the changes of the spectral composition (UV-B to far-red) and light direction and duration (photoperiod). Regardless of the fact photoreceptors can be found throughout the plant, in most cases, the site of light perception is the same as that where responses to the light stimulus take place. Nevertheless, there are examples where the site of light response is distant from the location of light perception, such as those with FTs responsible for floral transition and LD responses [31]. Photoreceptors are divided according to its wavelength sensing. They are named phytochromes in response to red/far-red (R/FR) light; cryptochromes, phototropins, and zeitlupes when sensing blue/UV-A light; and UV-B photoreceptors (UVR8) when responding to UV-B (Figure 1) [32]. Once photoactivated, photoreceptors interact with other players involved in light responses, such as ELONGATED HYPOCOTYL 5 (HY5), a photomorphogenesis promoting transcription factor, and the *CONSTITUTIVE PHOTOMORPHOGENIC 1* (*COP1*) - SUPPRESSOR OF PHYA-105 (SPA) complex, among others. COP1, a RING E3 ubiquitin ligase, forms a complex with SPA protein family (SPA1-SPA4), targeting key regulators for degradation [33]. Especially for SPA2, its action is rapidly repressed due to its degradation promoted by light-activated phytochrome photoreceptors but not by cryptochromes. The degradation of SPA2 is dependent on COP1′s ubiquitin role, meaning that the light regulation of COP1-SPA complex is achieved through the degradation of SPA2 protein but not of COP1 [34]. During the night, the COP1-SPA complex is no longer degraded by light-activated photoreceptors, enabling the target of HY5 and CO for ubiquitination and degradation with consequent suppression of photomorphogenesis and flowering, respectively [35]. DE-ETIOLATED 1 (DET1) is a protein that interacts with COP10 and DAMAGED DNA BINDING PROTEIN 1 (DDB1), establishing a complex that assists COP1 mediating the degradation of transcription factors responsible for light responses such as HY5, HY5 HOMOLOGUE (HYH), LONG AFTER FAR-RED LIGHT1 (LAF1), and LONG HYPOCOTYL IN FAR-RED1 (HFR1) [36]. In sum, COP1 and DET1 belong to the regulatory ubiquitin-mediated proteolic degradation machinery responsible for the inhibition of photomorphogenesis during the dark period.

The diverse photomorphogenic responses of plants, such as seed germination, de-etiolation, and circadian clock regulation, are, to a great extent, associated with light perception by photoreceptor system [37]. Photoreceptors act through the presence of specific photoreceptive chromophores [38,39], and are the gatekeepers for different types of light sensing which are responsible for regulating a high number of physiologically and developmental downstream signaling pathways [40].

## 2. Introduction to Photoreceptors

### 2.1. Phytochromes

It is possible to identify five photochromes in Arabidopsis: phyA, phyB, phyC, phyD, and phyE [41,42]. They are involved in several life cycle functions, including germination, de-etiolation, stomata development, flowering transition, senescence, and shade avoidance [43,44]. Phys are synthesized as a biologically inactive (Pr) form that is converted to an active (Pfr) form after red light absorption. This latter form is rapidly converted back to the inactive Pr state upon far-red light irradiation or slowly by dark reversion [45]. Pfr leads to the inhibition of several PHYTOCHROME INTERACTING FACTORs (PIFs) either by triggering their proteasome-mediated degradation or through their phosphorylation [44] (Figure 1a). The ubiquitin E3 ligase complex (COP1/SPA/FUS) is also inhibited by Pfr, enabling the stabilization of the transcription factors required for light-growth development such as HY5 [46]. In the dark, upon Pfr form spontaneously change back to the Pr form after phosphorylation, it acquires the capacity to enter the nucleus and establish subnuclear foci called photobodies [47]. These are heterogeneous, meaning that they can change composition throughout the day [48,49]. Photobodies act upstream of PIFs by promoting their phosphorylation and targeting them for degradation via ubiquitination [50,51].

### 2.2. Cryptochromes

Arabidopsis has two cryptochromes, Cry1 and Cry2 [52], both present in the nucleus. When activated by blue light, Cry1 migrates to cytosol, while Cry2 remains in the nucleus [53]. Cry1 plays a predominant function during de-etiolation, and Cry2 displays its main role in the photoperiodic control of flowering [54]. The light-activated Cry1 binds to SPA1, leading to the inhibition of the COP1-SPA1 E3 ubiquitin ligase, which, consequently, gives rise to an accumulation of HY5 and CO transcription factors, promoting de-etiolation and flowering, respectively [55,56] (Figure 1b). Regarding Cry2, once activated by blue light, it interacts with CRY-INTERACTING bHLH (CIB1, 2, 4, 5), directly activating the transcription factor *FLOWERING LOCUS T* (*FT*) in order to promote flowering [57]. The other CRY-interacting protein is the SUPRESSOR OF PHYTOCHROME A-105 1 (SPA1), which is responsible for positive regulation of the E3 ubiquitin ligase COP1 [55,58], leading to an inhibition of COP1 dependent degradation of transcriptional regulators such as bZIP transcription factor HYH and bHLH transcription factor HFR1 [56].

### 2.3. Phototropins

The photoreceptors that perceived UV-A and blue light are called phototropins (phots). Arabidopsis genome holds two: PHOT1 and PHOT2 [59]. The blue light is sensed by two flavin monoclueotide (FMN) chromophore-binding light oxygen voltage (LOV1 and LOV2) domains [60]. The conformational change promoted by light leads to autophosphorylation of the residues located both in the sensory and kinase domains [61,62]. Blue light triggers both phots but in different manners: It stimulates *PHOT2* and downregulates *PHOT1* [63]. These two genes are responsible for distinct mechanisms, such as phototropism and leaf positioning [64], the opening of stomas, and the accumulation of chloroplasts [65]. The activated phototropins bind to NONPHOTOTROPIC HYPOCOTYL 3 (NPH3) and ROOT PHOTOTROPISM2 (RPT2), initiating the phot signaling [66].

### 2.4. Zeitlupe Family

The zeitlupe family of LOV UV-A/blue light photoreceptors comprise ZEITLUPE (ZTL), FLAVIN-BINDING, KELCH REPEAT, F-BOX (FKF1), and LOV KELCH PROTEIN2 (LKP2) proteins [60]. These proteins contain only one LOV domain, followed by an F-box and six KELCH repeats [67]. Zeitlupes form SCF E3 ubiquitin ligase complexes, which directly control light-mediated protein degradation [21]. This activity has an important role in the photoperiodic control of floral transition [65], circadian oscillator regulation, and hypocotyl elongation [68].

The mechanism of action of ZTL, LKP2, and FKF1 is through the ubiquitin-mediated degradation of clock elements such as TOC1 and PRR5 [67,69]. The interaction of FKF1, ZTL, and LKP2 with GI is induced by blue light (Figure 1c). The FKF1-GI complex interacts with CDF1 (a repressor of CO and FT expression), leading to its degradation by the ubiquitin-proteasome system with subsequent expression of *CO* [22] and *FT*. The KELCH repeats are the regions responsible for substrate proteins recognition for ubiquitination [70], and both FKF1 and LKP2 are able to interact with CDF1 [71]. In sum, the light activation of the LOV domain in the zeitlupe family members promotes an activation with GI and modification of its ubiquitin E3 ligase activity. This way, the GI-FKF1 complex actively triggers the *CO* repressor for degradation. With this repression lifted, *CO* is expressed under light, modulating flowering. The ZTL-GI complex, on the other hand, when regulated by light, limits the ZTL degradation capacity of this target proteins, controlling the circadian clock response [59].

### 2.5. UVR8

The UV RESISTANT LOCUS 8 (UVR8) photoreceptor is on the front line of UV-B stress-induced responses and triggers a wide range of changes in gene expression, leading to morphological adaptions and the production of flavonols that act as UV-B protective shields [72]. It also mediates phototropic bending, stomatal movement, and circadian clock cascade gene activation [73,74]. The inactive form of UVR8 is a homodimer but is converted to an active monomer after absorbing UV-B [75]. In its active monomeric form, UVR8 interacts with E3 ubiquitin ligase COP1 in the nucleus [76], leading to the expression and stabilization of HY5 and HY5 HOMOLOGUE (HYH), which, in turn, bind to the promotor of several UV-B responsive genes [77] (Figure 1d), such as those involved in the flavonoid synthesis, and chalcone synthase [78,79], which is important in abiotic and biotic stress responses. When plants are subjected to low UV-light intensities for short periods of time, their actions beneficially enhance their resistance to pathogens and herbivores [80] but increase UV-light exposure, stimulates the generation of free radicals detrimental for DNA, proteins, lipids, chloroplasts, and photosynthetic pigments, hence impairing yield [81].

## 3. Plant Development under Light

### 3.1. Mechanisms Regulated by the Action of Light

Plants can sense light signals through photoreceptors, which are capable of transmitting those signals into a light-signaling response cascade [82]. The bHLH transcription factors PHYTOCHROME-INTERACTING FACTORs (PIFs), acting downstream of phytochromes, show a repressive action on photomorphogenesis during darkness or, in other words, promote skotomorphogenesis growth in the absence of light [44]. The *PIFs* family comprises *PIF1*/*PIF3*-*LIKE5* (*PIL5*), *PIF3*, *PIF4*, *PIF5*/*PIL6*, and *PIF6*/*PIL2* [42]. Under light conditions, active phytochromes interact with PIFs, resulting in phosphorylation and further PIFs degradation by proteasome, resulting in the photomorphogenesis responses. Recent findings on the regulation of PIF3, responsible for hypocotyl elongation, cotyledon expansion, and chloroplast development, have shown that new factors are involved in the degradation of PIF3 by the SKP1-CUL1-F-box protein complex (SCF), EIN3-BINDING F BOX PROTEINs (EBFs) 1 and 2, E3 ubiquitin LIGHT-RESPONSE BRIC-A-BRACK/TRAMTRACK/BROAD (LRB), and Photoregulatory Protein Kinases (PPKs) [83,84,85]. Regardless of PIF3’s ability to bind EBFs under light or dark conditions, the recruitment of PIF3-EBFs to the SCF^EBF1/2^ only occurs under light and after phosphorylation of PIF3 [84]. LRB, on the other hand, only targets phyB and PIF3 under high-light environments without affecting the stability of phyB [83]. PIF3 and phyB also interact with PPKs in a light-dependent fashion, which is required for PIF3′s phosphorylation [85]. With the migration of phyB to the nucleus upon activation by light, it binds to PIF3, followed by the activation of PPK with consequent phosphorylation of PIF3. The phosphorylated PIF3 form is recognized by the LRB E3 ligases, resulting in the degradation by ubiquitination of both PIF3 and phyB [83,85].

Other PIFs are also involved in the control of germination. Specifically, PIF1 represses germination and de-etiolation [86]. This negative regulation is achieved through the action of the KELCH F-BOX protein COLD TEMPERATURE GERMINATION (CTG10), which recognizes, binds, and destabilizes PIF1, hence stimulating the completion of germination and seedling de-etiolation. This activation also shows a feedback loop where PIF1 downregulates the transcription of *CTG10* [87]. Light quality also affects seed germination through the regulation of gibberelins (GA) and abscisic acid (ABA) levels. Whereas GA induces germination, ABA inhibits it. Valstij et al. [88] showed that under far-red conditions (usually shade environments), the gene *MOTHER-OF-FT-AND-TFL1* (*MFT*) represses germination by regulating both ABA and GA signaling pathways. In the dark, PIF1 promotes the expression of *ABA-INSENSITIVE 5* (*ABI5*), *DELLA-encoding GA-INSENSITIVE* (*GAI*), and *REPRESSOR-OF-GA1* (*RGA*) [89]. The control of ABA and GA accumulation by PIF1 is also exerted by promoting the activation of *SOMNUS* (*SOM*), which is responsible for the increasing of ABA and decreasing of GA levels [90].

### 3.2. Flowering Control by CO/FT Regulation

Flowering is a fundamental phase of plant’s development and is of great importance when dealing with species that are the sources of seeds which are essential for feeding purposes. Flowering regulation depends on internal and external signals such as temperature and daylength, which are converted into the regulation of two major floral integrator genes: *FT* and *CO* [91]. *FT* is upregulated by the protein CO in the photoperiodic pathway, which is initiated by the photoreceptors CRY1, CRY2, and FKF1 [27]. 

Day length must reach a threshold in order to promote the stabilization of the CO protein. Only under LD, CO is not degraded, picking its accumulation during the night [92,93]. CO, alone or combined with other factors, activates the transcription of *FT* gene (Figure 2). The flowering time is dependent on the amount of FT protein present in the apical meristems, which was previously transported from leaves, where the *FT* gene is regulated by a high number of factors. CO presents a tight transcriptional and post-transcriptional regulation, which is important to measure the day length and calculate when to promote flowering. The circadian clock FKF1, GI, and CDF are major factors in the regulation of CO [71]. In the case of CDF1 protein, it directly binds to the *CO* promoter and represses its expression, which results in a reduced level of *CO* transcripts in the morning [94] (Figure 2a). Conversely, *CDF1* is positively regulated by CCA1 and LFY at dawn and negatively regulated by PRR9, PRR7, and PRR5 in the afternoon [95].

Under LD days, the FKF1-GI complex degrades CDF proteins, preventing its repression on the *CO* promoter [96]. As days shorten, *FKF1* and *GI* transcription levels no longer pick at the same time. Hence, the FKF1-GI complex cannot be established, and CDF proteins are not degraded, remaining present on the *CO* promoter. In this way, *CO* transcription is prevented, which, in turn, does not activate *FT* expression during the day under SD [93]. Other genes are involved in the regulation of the CO-FT pathway. ZTL and LKP2 interact with FKF1 and GI to help in the destabilization of the CDF2 protein [22]. The bHLH (basic helix-loop-helix) transcription factors FLOWERING BHLH1 (FBH1), FBH2, FBH3, and FBH4 promote the activation of *CO* once the CDF repressor is removed from the *CO* promoter region [97]. Albeit *CO* transcripts accumulate at night both under LD and SD, their effects on the *FT* gene only take place at dusk under LD. CO proteins are degraded by COP1-SPA1 complex during the night (Figure 2a) [98], while during the first hours of the day, CO degradation is carried out by the HIGH EXPRESSION OF OSMOTICALLY RESPONSIVE GENE1 (HOS1), a RING finger containing E3 ubiquitin ligase [99]. PHYB also mediates CO protein degradation in the morning, contrarily to PHYA, which stabilizes CO later in the day [100]. CRY1 and CRY2 are another group of photoreceptors involved in CO protein regulation [55,101] with CRY2-SPA1 upon light activation, promoting the interaction between CRY2 and COP1 and inhibiting the capacity that COP1 has to degrade CO (Figure 2a) [55].

The transcriptional activation/inhibition of *FT* is the final approach for flowering regulation initiated by light sensing. The heterodimer comprising FLC and SHORT VEGETATIVE PHASE (SVP) directly suppress *FT* and *SOC1* expression [102]. *FT* transcription is also inhibited by the FLC-FLM (FLOWERWING LOCUS M)-MAFs (MADS AFFECTING FLOWERING) complex [103] and by the Polycomb group complex (PRC 1-like complex) composed by the EMBRYONIC C FLOWER 1 (EMF1), LHP1 and histone E3 lysine-4 [92]. *FT* is negatively regulated by three more transcription factors: SCHLAFMÜTZE (SMZ), TEMPRANILLO1 (TEM1), and TEM2 (Figure 2b) [104,105]. As *FT* activators, one finds CIBs and CO proteins. CIBs interact directly with *FT* promoter, activating its expression, but the CIB1 function is restricted from late afternoon to early night and forms a complex with CRY2 after photoexcitement by blue light (Figure 2b). In the absence of blue light, CIB1 is targeted for degradation [106], while its stabilization is mediated by ZTL and LKP2 under blue light with further *FT* expression [101].

## 4. Conclusions

The influence of light on plant development, curiously, starts with the lack of light during seed germination until soil surface is reached. At this time, plant development suffers a transition from the heterotrophic growth, where maternally deposited energy reserves are used to fuel an autotrophic photosynthesis-dependent growth. As soon seedlings are exposed to light, fully functional chloroplasts develop, enabling photosynthesis to take place in reactions dependent on light-absorbing pigment molecules present in the thylakoids. Here, light energy is converted to chemical energy, which involves a series of chemical reactions known as the light-dependent reactions. Plants also possess five classes of photoreceptors that are differentially photoactivated based on the type of wavelengths they absorb (from far-red to UV-B). This distinguished capacity of the different photoreceptors to absorb in narrow zone of the solar spectrum allows a precise physiological response in the face of diverse signals from light, such as intensity, quality, and duration. Therefore, throughout plant’s lifespan, we find mechanisms such as the circadian clock, which can be summarized as endogenous genetic pathways that allow plants to anticipate and prepare for environmental daily and seasonal changes. Among the circadian clock, we can find biological processes such as hypocotyl elongation, photosynthesis, stomatal opening, leaf movement, cell cycle progression, and flowering.

Photomorphogenesis, also defined as light-regulated plant development molecular signaling intermediates, are formed in response to the activation of photoreceptors by light. The photomorphogenic physiological responses are germination, de-etiolation, shade avoidance, circadian rhythm, and flowering. Flowering was singled out in this review given its importance for plant multiplication and seed production, which is vital for animal feeding.

The present work intended to show, in a relatively brief way, that the balance between light-promoter development and growth under darkness is delicate and complex, with the capacity to influence all aspects of plant’s life. If we consider the wider view of all life on Earth, having plants as the base of the food chain, it is easy to understand the importance of having a deep knowledge of how light perception is mediated by all photosynthetic organisms.

## Figures and Tables

**Figure 1 plants-09-00894-f001:**
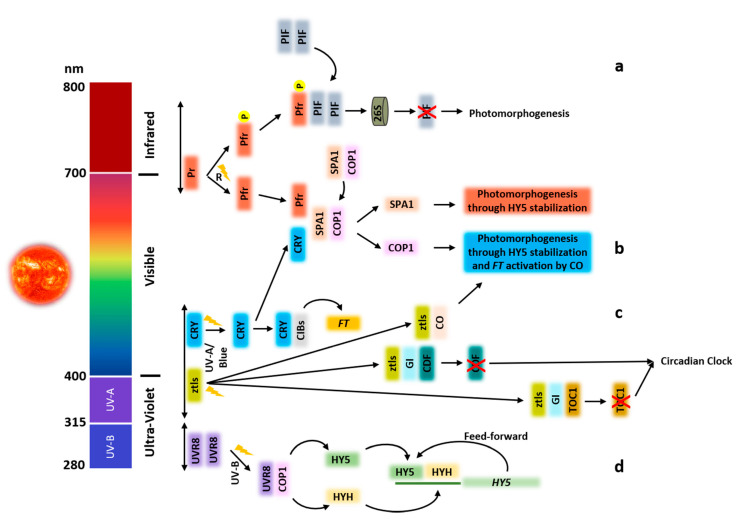
Proposed explanatory model of plants photoreceptors activity upon activated by light. (**a**) The biological active Pfr form of the phytochrome is activated by red light (R). The interaction between Pfr and PIFs triggers the rapid phosphorylation of PIFs entering protein degradation via the ubiquitin 26S proteasome pathway promoting photomorphogenesis. Pfr also induces COP1-SPA1 complex dissociation. All these pathways lead to the accumulation of major transcription factors (TFs) triggering photomorphogenesis such as HY5 (**b**) Two mechanisms are represented for cryptochrome signal transduction. Transcription regulation of light-activated Cry after interaction with TF CIB1 and its relatives (CIBs) leads to the activation of FT transcription, promoting floral initiation. The other mechanisms involve the interaction of cryptochromes with SPA1 proteins to suppress SPA1 activation of COP1 activity necessary for the degradation of HY5, HYH, CO, and other transcription regulators promoting photomorphogenesis. (**c**) ztls are activated by UV-A/blue light. Once activated, they interact with GI, resulting in the stabilization of these receptors, enabling them to act as E3 ligases which target the transcriptional regulators CDF and TOC1 for degradation. In addition, light-activated FKF1 interacts with and stabilizes CO. (**d**) The inactive dimeric form of UVR8 is activated to the monomeric form by UV-B, which interacts with the E3 ubiquitin ligase COP1. With the suppression of COP1, HY5 and HYH are stabilized. These two proteins, in a feed-forward loop, bind to the *HY5* promoter region with further activation of transcription.

**Figure 2 plants-09-00894-f002:**
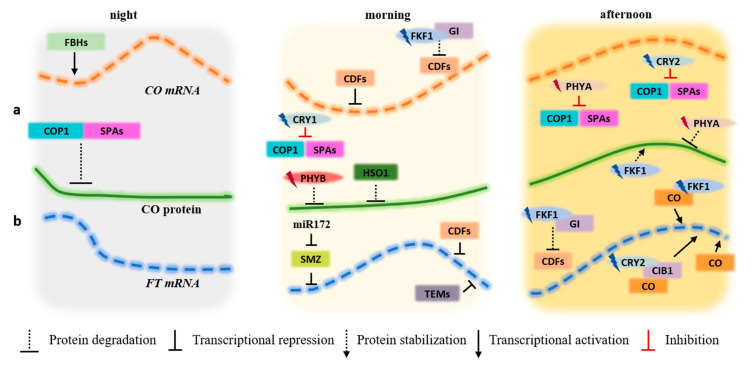
CO and FT photoperiodic regulation under LD during the night, morning, and afternoon periods. (**a**) Transcriptional (*CO* mRNA) and post-translational (CO protein) regulation. (**b**) Transcriptional regulation of FT (FT mRNA). At night, the FBHs (bHLH transcription factors) bind to the *CO* promoter, activating its transcription. In the morning, the presence of high amounts of CDF proteins on the *CO* and *FT* promoters repress the transcription of both genes. As afternoon is reached, FKF1 is photoactivated by blue light. At this point, it forms a complex with GI, promoting the degradation of the CDFs proteins which bind to the CO and FT promoters, freeing these promoter regions for binding to other gene activators. At the protein regulation level, the COP1-SPAs complex degrades CO protein during the night hours. When CRY1 and CRY2 are photoexcited by blue light, they acquire the capacity to interact with COP1 and SPAs, inhibiting the COP1-SPAs complex activity and increasing the stability of CO. PHYA also inhibits the COP1-SPAs complex. During the first hours of the day, HOS1, ubiquitin ligase, and red light activated PHYB mediate CO degradation. By the time CO picks late in the afternoon, blue light-activated FKF1 and far-red photo-activated PHYA stabilize CO. The amount of CO protein in the morning is low due to interactions with other proteins that prevents the CO-dependent activation of *FT*. Throughout the day, as CO becomes stable upon interaction with blue light-activated FKF1 and CRY2, activates the transcription of *FT* gene by directly binding to the *FT* promoter. The blue light-activated form of CRY2 also interacts with CIB1 and directly binds to the *FT* promoter, activating the transcription of the gene late in the afternoon. miR172 is a negative regulator of *SMZ* and SMZ protein and is a powerful repressor of FT expression under LD. TEMs transcription factors (TEM1 and TEM2) bind to FT promoter, inhibiting its transcription in the morning.

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
