# Peer review of "Distinct Responses to Light in Plants"

_plants, 2020, doi:10.3390/plants9070894_

Round 1
Reviewer 1 Report
The author summarized some progresses in light signal transduction, involving photoreceptors, light-regulated developmental process (e.g. flowering) etc. The manuscript introduces the regulation network among key players in light signal transduction as well as flowering, such as phytochromes, COP/DET/FUS, CO/FT and circadian clock, which will provide an overview of how plant senses light to the readers. However, I have several major concerns to be addressed, and I think this will make the manuscript more accurate and inclusive:
1. The author organizes the manuscript in a way as: 1. Introduction. 2 phytochromes. 3 Cryptochromes… 7 Growth under light… Obviously, these titles are not at the same level, and re-organization of the titles with proper subtitles will make the structure of the manuscript more reasonable and clear, unless there is specific regulations that is stated by the journal. A suggestion would be:
1 Overview of light transduction scheme.
2 Introduction to photoreceptors. 2.1 phytochromes; 2.2 cryptochromes etc.
3 Plant development under light: 3.1 Growth under light; 3.2 flowering control etc.
4 Plant growth without light…
2. When talking about the phosphorylation and degradation of phytochromes and PIF proteins, I think people have made major advances in recent years, and following original studies should be cited and mentioned to make this part more complete: Ni et al., plant cell 2013 DOI: 10.1105/tpc.113.112342 ; Ni et al., science 2014 DOI: 10.1126/science.1250778; Ni et al., Nature communications 2017 https://www.nature.com/articles/ncomms15236; Dong et al., current biology 2017 https://doi.org/10.1016/j.cub.2017.06.062. These studies promoted the understanding of post-translational regulation of phyB and PIF3, which will provide more accurate and up-to-date information about how plant senses light.
3. Line 126-129, please cite the exact reference(s) that shows light-activated photoreceptors degrade COP1 under light but not in the dark. Line 130-131, as far as I know, the total abundance of COP1 and DET1 are not affected by light signal; the literature cited [#36] in the manuscript highlighted the translocation of COP1 between nucleus and cytosol instead of total protein level reduction, a more specific definition of “COP1 and DET1 are reduced” was needed here.
4. Line 101&line 142-143 & line 160-161& line 186, something may be missing in these sentences, which makes it hard to understand the meaning.
5. Line 351-352, “by degrading transcription factors via…” there are too many transcription factors in planta, and it should be pointed out which class of transcription factors are here.
6. Line 104, “it is expressed via CO…” maybe change to “its expression relies on CO…”?
7. Line 138, as the key point of this review is how plant senses light, I think maybe mentioning “dark reversion” is better than “thermal reversion” here about the transformation of phytochromes.
8. Please go through all the citation styles in the text, it is not consistent; many places used XXX et al., XXXX.
9. Typos: line 212, “UV8”, UVR8; line 227, “PHOTOCHROME-INTERACTING FACTORs”, PHYTOCHROME-INTERACTING FACTORs;
Author Response
Response to Review 1
I would like to sincerely thank the reviewer for a thorough and comprehensive review of the manuscript. Driving my attention to new findings and new studies was precious and I think the quality of the work has improved substantially. All comments were accepted and included in the text.
I have created two documents. One showing all the alterations made on the text and a second one that is a “cleaned” version of the first one. In this second one is still possible to see text box indicating the suggestions made from each of the reviewers. Hopefully, will help the next review.
I have removed the section “growth under dark” because it was a mess and did not make sense. Instead, I extended the last paragraph as a summary of what explained throughout the review.
- The author organizes the manuscript in a way as: 1. Introduction. 2 phytochromes. 3 Cryptochromes… 7 Growth under light… Obviously, these titles are not at the same level, and re-organization of the titles with proper subtitles will make the structure of the manuscript more reasonable and clear, unless there is specific regulations that is stated by the journal. A suggestion would be:
1 Overview of light transduction scheme.
2 Introduction to photoreceptors. 2.1 phytochromes; 2.2 cryptochromes etc.
3 Plant development under light: 3.1 Growth under light; 3.2 flowering control etc.
4 Plant growth without light…
The sub-division of the text already reflects the reviewer suggestion.
- When talking about the phosphorylation and degradation of phytochromes and PIF proteins, I think people have made major advances in recent years, and following original studies should be cited and mentioned to make this part more complete: Ni et al., plant cell 2013 DOI: 10.1105/tpc.113.112342 ; Ni et al., science 2014 DOI: 10.1126/science.1250778; Ni et al., Nature communications 2017 https://www.nature.com/articles/ncomms15236; Dong et al., current biology 2017 https://doi.org/10.1016/j.cub.2017.06.062. These studies promoted the understanding of post-translational regulation of phyB and PIF3, which will provide more accurate and up-to-date information about how plant senses light.
The research work on these publications are very important and upon including them in the manuscript, ideas are now more complete and clearer.
- Line 126-129, please cite the exact reference(s) that shows light-activated photoreceptors degrade COP1 under light but not in the dark. Line 130-131, as far as I know, the total abundance of COP1 and DET1 are not affected by light signal; the literature cited [#36] in the manuscript highlighted the translocation of COP1 between nucleus and cytosol instead of total protein level reduction, a more specific definition of “COP1 and DET1 are reduced” was needed here.
The reference was corrected and the COP1 mechanism clarified.
- Line 101&line 142-143 & line 160-161& line 186, something may be missing in these sentences, which makes it hard to understand the meaning.
I am so sorry for these nonsense sentences. Hopefully, they read better now.
- Line 351-352, “by degrading transcription factors via…” there are too many transcription factors in planta, and it should be pointed out which class of transcription factors are here.
The different TFs are now enumerated in the text.
- Line 104, “it is expressed via CO…” maybe change to “its expression relies on CO…”?
Has been changed
- Line 138, as the key point of this review is how plant senses light, I think maybe mentioning “dark reversion” is better than “thermal reversion” here about the transformation of phytochromes.
Has been changed
- Please go through all the citation styles in the text, it is not consistent; many places used XXX et al., XXXX.
Again, I am sorry for this lack of uniformity. I did convert the references to Plants’s format right before submitting the manuscript but apparently, the software did not convert all the references even though they have been added. At this point, I must address this issue with the journal.
- Typos: line 212, “UV8”, UVR8; line 227, “PHOTOCHROME-INTERACTING FACTORs”, PHYTOCHROME-INTERACTING FACTORs
It has been corrected.

Reviewer 2 Report
Review of the manuscript: How plants sense light
The manuscript deals with the interesting and widely discussed problem of plant reaction to light. The author describe in a very brief way the use of light in the process of photosynthesis (maybe even too briefly) and the process of photomorphogenesis much broader. Flowering regulation is discussed in detail.
The paper lacks consistency and a systematic approach, e.g. the construction of some photoreceptors is discussed, while others are not. Reading the text, you can feel the chaos of thoughts - the desire to convey as much information as possible in several sentences. For example, the structure of individual photoreceptors and their functions and regulation mechanisms can be clearly discussed. Maybe the text division into light perception and light signaling sections will be more reader-friendly.
Detailed comments:
- keywords - require supplementation
- line 30: earth’s orbit around the sun providing a diurnal rhythm of day and night; the movement of the earth around its axis determines the change of day and night, and the movement of the earth around the sun - it changes the seasons
- line 38: which involves a set of environmental adaptations; not adaptation but adjustments, adaptations are inheritable
- some references in the text are given by numbers and others are given names - please unify
- line 48-49 - sentence not clear, what author mean plant's abortion?
- line 53-54: photosynthetic pigments that absorbs light at a narrow range of the solar spectrum.; Photosynthetic pigments absorb throughout the PAR spectrum, only the absorption peaks are narrow
- abbreviations should be explained during the first use, e.g. MYB in line 69
- The author uses many old literature items, published before 2010, maybe it is worth exchanging for newer items? And the use of item from 1984 (85) and about phytoplankton seems not justified
- In the Introduction section there is a broad description of the regulation of the flowering process - which is later discussed in the next section - this introduction should be shortened
In the Introduction, the authors omit other important processes regulated by light such as germination, growth - sentence: Plants make use of a vast number of different types of photoreceptors so that they can accurately detect the changes of the spectral composition (UV-B to far-red), light direction and duration (photoperiod). is quite similar to sentence: Plants can perceive changes in light quality, intensity, direction, and duration through their different types of photoreceptors. It is unnecessary repetition
- line 213 should be UVR8
- line 226 why author write four? there are much more photoreceptors.
- line 231 should be skotomorphogenesis
- The Growth under light chapter is chaotic and does not fully present the subject
- line 261 is 'plant' better will be 'stem'
- line 271 vernalization is not a signal, temperature is a signal
- figure 2 looks quite similar to fig 2.7 in The Light Awakens! Sensing Light and Darkness, Eros Kharshiing, Yellamaraju Sreelakshmi, and Rameshwar Sharma, Sensory Biology of Plants, 2019, what is the upgrade?
- line 340-342 not clear
- the last paragraph should be expanded to summarize and clearly separated from the previous section
- there is lack of: Author Contributions, Funding, Acknowledgments, Conflicts of Interest
21 In some references items there is lack of data, e.g. journal title (e.g. 58)
The topic is very interesting, but the author should divide the text into clear and logical subchapters. In such a wide topic, you can not ignore some aspects of plant photobiology. Or you should change the title.
The work requires significant improvements, in its current form cannot be accepted.
Author Response
Response to reviewer 2
I would like to sincerely thank the reviewer for a thorough and comprehensive review of the manuscript. Driving my attention to new findings and new studies was precious and I think the quality of the work has improved substantially. All comments were accepted and included in the text.
I have created two documents. One showing all the alterations made on the text and a second one that is a “cleaned” version of the first one. In this second one is still possible to see text box indicating the suggestions made from each of the reviewers. Hopefully, will help the next review.
keywords - require supplementation
more keywords were added.
line 30: earth’s orbit around the sun providing a diurnal rhythm of day and night; the movement of the earth around its axis determines the change of day and night, and the movement of the earth around the sun - it changes the seasons
I am sorry for the lack of precision on this matter. The sentence was changed in order to correct this.
line 38: which involves a set of environmental adaptations; not adaptation but adjustments, adaptations are inheritable
The change was made.
some references in the text are given by numbers and others are given names - please unify
I am sorry for this lack of uniformity. I did convert the references to Plants’s format right before submitting the manuscript but apparently, the software did not convert all the references even though they have been added. At this point, I must address this issue with the journal.
line 48-49 - sentence not clear, what author mean plant's abortion?
I have removed the sentence. It didn’t add anything.
line 53-54: photosynthetic pigments that absorbs light at a narrow range of the solar spectrum.; Photosynthetic pigments absorb throughout the PAR spectrum, only the absorption peaks are narrow
Thank you for calling out for the lack of precision of this statement. It is now corrected.
abbreviations should be explained during the first use, e.g. MYB in line 69
The MYB domain is now described with more detail.
The author uses many old literature items, published before 2010, maybe it is worth exchanging for newer items? And the use of item from 1984 (85) and about phytoplankton seems not justified
Corrected
In the Introduction section there is a broad description of the regulation of the flowering process - which is later discussed in the next section - this introduction should be shortened
In the Introduction, the authors omit other important processes regulated by light such as germination, growth
I have added a paragraph about germination in point 3. 1. “Growth Mechanisms regulated by the action of light”
sentence: Plants make use of a vast number of different types of photoreceptors so that they can accurately detect the changes of the spectral composition (UV-B to far-red), light direction and duration (photoperiod). is quite similar to sentence: Plants can perceive changes in light quality, intensity, direction, and duration through their different types of photoreceptors. It is unnecessary repetition
Corrected
line 213 should be UVR8
Corrected
line 226 why author write four? there are much more photoreceptors.
Corrected
line 231 should be skotomorphogenesis
You are right, it is now corrected.
The Growth under light chapter is chaotic and does not fully present the subject
The chapter names were re-organized. I also added a paragraph about germination widening this way the description light effects on plants.
line 261 is 'plant' better will be 'stem'
Corrected
line 271 vernalization is not a signal, temperature is a signal
Corrected
figure 2 looks quite similar to fig 2.7 in The Light Awakens! Sensing Light and Darkness, Eros Kharshiing, Yellamaraju Sreelakshmi, and Rameshwar Sharma, Sensory Biology of Plants, 2019, what is the upgrade?
Figure 2 was re-drawn and it contain some more information especially when it comes to the representation of the genes involved in FT regulation. I must admit I struggle in coming up with a scheme that does not resembles others without compromising important information. Hope this one can be seen as something new.
line 340-342 not clear
corrected
the last paragraph should be expanded to summarize and clearly separated from the previous section
I have removed the section “growth under dark” because it was a mess and did not make sense. Instead, I extended the last paragraph.
there is lack of: Author Contributions, Funding, Acknowledgments, Conflicts of Interest
Corrected
21 In some references items there is lack of data, e.g. journal title (e.g. 58)
Corrected. All new references were introduced.
The topic is very interesting, but the author should divide the text into clear and logical subchapters. In such a wide topic, you can not ignore some aspects of plant photobiology. Or you should change the title.
I have changed the title

Reviewer 3 Report
Dear author
Thanks a lot for your work. Please, find below a few comments of mine on the work
- Well written, nice storytelling and easy to read
- Please, double-check if all gene names, etc., that have to be italic, are actually italic
- I suggest minimalizing the figure design. The figures contain a lot of information, if they contain more information via the design, it can become quite overwhelming to process. E.g.:
-
- Reduce colours
- Use defined boxes, instead of one with blurring frames
- Use same font size as text (at least not bigger)
- …
- Add further figures to the other section, if possible
Author Response
Response to Review 3
I would like to sincerely thank the reviewer for a thorough and comprehensive review of the manuscript. I think the quality of the work has improved substantially. All comments were accepted and included in the text and figures.
I have created two documents. One showing all the alterations made on the text and a second one that is a “cleaned” version of the first one. In this second one is still possible to see text box indicating the suggestions made from each of the reviewers. Hopefully, will help the next review.
I have removed the section “growth under dark” because it was a mess and did not make sense. Instead, I extended the last paragraph as a summary of what explained throughout the review.
Please, double-check if all gene names, etc., that have to be italic, are actually italic
I have checked and add a few more genes. Regardless, I will check again right before the final proof to be absolutely sure.
I suggest minimalizing the figure design. The figures contain a lot of information, if they contain more information via the design, it can become quite overwhelming to process. E.g.:
Reduce colours
Use defined boxes, instead of one with blurring frames
Use same font size as text (at least not bigger)
I have changed figure 2 to show a better design in the text. I kept the blurring boxes because when I removed this feature, visually, the picture became quite heavy.
…
Add further figures to the other section, if possible
For the sake of space, I condensed information from the first sections into Fig. 1. Still hope is readable without too much effort.

Round 2
Reviewer 1 Report
The manuscript has been significantly improved, and all my concerns are adequately addressed by the author.
Author Response
Reply to reviewer 1
I would like to thank the time reviewer 1 took to read and for all the comments that that allowed me to significantly improve the manuscript.
Best regards
rita teixeira

Reviewer 2 Report
The Author has accepted all suggestions
and significantly improved the manuscript
divided the text into sections and is better to read now
However, the topic is very extensive and the work focuses on selected aspects
In my opinion, the text from line 339 should be separated as a summary and not be part of the chapter about flowering
Author Contributions part should be deleted
After minor improvement could be accepted
Author Response
Reply to reviewer 2
I would like to thank the time reviewer 1 took to read and for all the comments that that allowed me to significantly improve the manuscript.
In my opinion, the text from line 339 should be separated as a summary and not be part of the chapter about flowering
All the text after line 339 is part of the final remarks and I have added “4. Conclusion” to completely separate this final part of the manuscript.
Author Contributions part should be deleted
Corrected
Best regards
rita teixeira
